# Comparison of Different Cervical Finish Lines of All-Ceramic Crowns on Primary Molars in Finite Element Analysis

**DOI:** 10.3390/ma13051094

**Published:** 2020-03-01

**Authors:** Chin-Yun Pan, Ting-Hsun Lan, Pao-Hsin Liu, Wan-Ru Fu

**Affiliations:** 1Division of Orthodontics, Department of Dentistry, Kaohsiung Medical University Hospital, Kaohsiung 80756, Taiwan; 2Division of Prosthodontics, Department of Dentistry, Kaohsiung Medical University Hospital, Kaohsiung 80756, Taiwan; tinghsun.lan@gmail.com; 3School of Dentistry, College of Dental Medicine, Kaohsiung Medical University, Kaohsiung 80708, Taiwan; 4Department of Biomedical Engineering I-Shou University, Kaohsiung 82442, Taiwan; phliu@isu.edu.tw; 5Department of Stomatology, Kaohsiung Veterans General Hospital, Kaohsiung 81362, Taiwan; pipi7322@msn.com

**Keywords:** deciduous teeth, stress analysis, all-ceramic crown, cervical finish line

## Abstract

This study aimed to conduct a stress analysis of four types of cervical finish lines in posterior all-ceramic crowns on the primary roots of molar teeth. Four different types of finish lines (shoulder 0.5 mm, feather-edged, chamfer 0.6 mm, and mini chamfer 0.4 mm) and two all-ceramic crown materials (zirconia and lithium disilicate) were used to construct eight finite element primary tooth models with full-coverage crowns. A load of 200 N was applied at two different loading angles (0° and 15°) so as to mimic children’s masticatory force and occlusal tendency. The maximum stress distribution from the three-dimensional finite element models was determined, and the main effect of each factor (loading type, material, and finish line types) was evaluated in terms of the stress values for all of the models. The results indicated that the loading type (90.25%) was the main factor influencing the maximum stress value of the primary root, and that the feather-edged margin showed the highest stress value (*p* = 0.002). In conclusion, shoulder and chamfer types of finish lines with a 0.4–0.6 mm thickness are recommended for deciduous tooth preparation, according to the biomechanical analysis.

## 1. Introduction

Dental caries in young children may affect their body weight, growth, and quality of life [1]. If the caries remain untreated, they can rapidly progress into cavitation, involve dental pulp, and lead to dental infection or dangerous fascial space involvement. Low-level caries of primary teeth can be treated by caries removal, cavity preparation, and cavity restoration using materials such as composite resins or glass ionomers [2,3]. However, teeth with large carious lesions or those subjected to pulpotomy/pulpectomy require full-coverage prostheses such as stainless steel crowns (SSCs) [4] and all-ceramic crowns [5,6].

Preformed metal crowns for deciduous molar teeth have been available since the 1950s [7,8]. SSCs are metal caps that are used by dentists to cover decayed primary teeth and to prevent further decay. Traditional SSCs allow dentists to trim, crimp, and shape in order to better adapt to the cervical portion of the primary teeth. Santamaría et al. [9], in their questionnaire-based study in Germany, reported that dentists generally find SSCs to be technically complex, esthetically displeasing to both parents and children (because of their metallic appearance), time-consuming, and unprofitable. Resin-faced SSCs have emerged as a better alternative to metal SSCs, as they are more esthetic [10,11]. However, they are more prone to fracturing, debonding, and wearing of the buccal surface [12,13]. Pediatric all-ceramic crowns were introduced in 2010, and have gained popularity because of their improved esthetics and durability [12,14,15,16]. These zirconia crowns are prefabricated [15] using computer-aided design/computer-aided manufacturing (CAD/CAM) systems [17]. Zirconia (ZrO_2_), also known as ceramic steel, has favorable esthetics for primary teeth and has excellent mechanical properties, which include a high elastic modulus (215 Gigapascal (GPa)) and high flexure strength (1000 Megapascal (MPa)). Aiem et al. [18] found that zirconia crowns exhibited better gingival health and fewer crown fractures than the other ceramic crowns. Lithium disilicate (Li_2_Si_2_O_5_) [19], a popular glass-ceramic material, was first used in 1998 and is named the Ivoclar Porcelain System (IPS) Empress 2 (Ivoclar Vivadent, Schaan, and Liechtenstein). IPS Empress 2 is a second-generation, heat-pressed lithium disilicate crystal embedded in a glassy matrix. Later, IPS e.max (Ivoclar Vivadent, Schaan, and Liechtenstein) entered the market using the heat-pressed CAD/CAM fabrication method, with a flexural strength (biaxial) reaching 400 ± 40 MPa [20] and a Young’s modulus of 95 ± 5 GPa [20]. Glass-ceramic is more transparent than zirconia and is easier to surface-etch for resin cement bonding. However, all-ceramic crowns cannot be manipulated with crimping or contouring, and may require a passive fit and extensive tooth reduction.

Prefabricated primary zirconia crowns are thicker than SSCs, with occlusal thicknesses varying from 1 to 2 mm [15]. Some data from Lin Jing et al. [21] showed that 2 mm occluso-cervical heights were crucial for retention. However, other research indicated that the minimum thickness of zirconia for natural teeth should be 0.5 mm [22]. Alghazzawi et al. [23] showed that CAD/CAM zirconia laminate veneers with a 0.3–0.5 mm reduction can undergo higher loads than glass-ceramic veneers. Lan et al. [24] suggested that monolithic zirconia crowns with a0.8 mm occlusal thickness could face operative errors and deviation in occlusal adjustment. Regarding glass-ceramics, the manufacturer of IPS e.max, for example, suggested a 1.5 mm reduction in the occlusal thickness and a 1.0 mm reduction in the margin thickness. Sasse et al. [25] showed that the 0.7–1.0 mm occlusal thickness of IPS e.max is appropriated in adult posterior dentitions. Almost all of the data concerning occlusal thickness showed that the tooth reduction volume of CAD/CAM all-ceramic crowns could be less than that of prefabricated primary zirconia crowns.

Proffit and Fields [26] studied the molar bite forces of adults and children, and found mean bite equivalent forces of 31.9 kg (mean age 26.9 years old) and 17.4 kg (mean age 9.3 years old), respectively, at a 2.5 mm opening, and forces of 35.6 kg and 15.5 kg, respectively, at a 6.0 mm opening, which confirmed that children have a lower bite force than adults. The mean size of deciduous teeth is smaller than that of permanent teeth, and the retention of an all-ceramic crown depends on the tooth preparation and cementation. For the preparation of all-ceramic crowns for primary teeth, the occlusal volume and cervical finish line types must be seriously considered [15]. Unfortunately, as of yet, we could not find a study about the margin design of all-ceramic crown of primary root.

The margins of primary, full-coverage crowns can be divided into four categories concerning cervical finish types [15,27,28], namely: feather-edged, shoulder, mini chamfer, and chamfer. The feather-edged type is used for SSCs and some prefabricated zirconia crowns, as suggested by the manufacturer [15], with the advantage of tooth preservation. For the CAD/CAM all-ceramic crowns, however, the intact margin of the prosthesis tests the ability of the milling machine. The chamfer and shoulder margins are easily milled, but they require removing more of the tooth structure.

The aim of this study was to compare the stress distribution of four different cervical finish lines of all-ceramic crowns made from two different materials on primary roots using finite element analysis (FEA).

## 2. Materials and Methods

### 2.1. Model Preparation

Four primary mandibular second molars were prepared to four finish lines from the Nissin primary tooth model (Nissin Dental Products Inc., Kyoto, Japan), as follows: shoulder (0.5 mm), feather-edged, chamfer (0.6 mm), and mini chamfer (0.4 mm). The mean occluso-cervical heights of the buccal and lingual surfaces were 3 mm, and the proximal occluso-cervical heights were 2.0 mm.

### 2.2. Finite Element Method (FEM)

Four primary tooth models with different finish lines and full-coverage crowns were constructed using finite element software (Pro/Engineer Wildfire 2.0; Parametric Technology Corporation, Needham). All of the models were combined through Boolean operations using CAD software (Pro/ENGINEER, Wildfire 2.0; Parametric Technology Corp., Boston, MA, USA). ANSYS was the FEM software used in this study (ANSYS Workbench 11, Canonsburg, PA, USA). The material properties are listed in Table 1 [20,29,30,31,32].

The materials used in the models were assumed to be homogeneous, isotropic, and linearly elastic. A three-dimensional finite element mesh of the shoulder-type model comprising 669,499 elements and 992,224 nodes was built using 10 node tetrahedral elements. Six degrees of freedom (DOF) were provided by the 10 node tetrahedral elements. Hence, the total number of DOF in the FEM was related to the real nodal numbers. The mesh convergence test of the element sensitivity was performed and set to 5% so as to identify a reliable numerical result for the FEM. 

For the contact connection between the all-ceramic crown and primary tooth, we defined the finite element analysis (FEA) as a non-separating connection without any frictional effect. Hence, a contact magnitude of 200 N and an applied oblique force of 200 N were set 15° from the long axis of the tooth [33]. All of the models were constrained in all directions at the nodes on the distal and mesial borders of the bone surface. The von Mises stress is defined as the beginning of the deformation of materials with non-direction, and showed a larger damage trend in this qualitative study [34].

### 2.3. Statistical Analysis

This study focused on the maximum von Mises stress values in primary teeth. Dar et al. [35] showed how computational effort could be minimized based on statistical methods; therefore, to reduce the complexity of the results, the main effect of three investigator factors (loading type, cervical finish line type, and crown material) on the mechanical response (stress) was analyzed based on statistical processes [36,37]. The data were compared using analysis of variance (ANOVA), followed by the Tukey–Kramer multiple comparison test. All of the statistical analyses were conducted using IBM, the International Business Machines Corporation, SPSS Statistics for Windows, version 20.0 (IBM Corp., Armonk, NY, USA). The analysis demonstrated the percentage contribution of each investigated factor to the sum of squares (TSS), and helped determine the factors that were minimizing stress on the primary roots. A statistical significance value in all cases was accepted as *p* < 0.05.

## 3. Results

As revealed in Figure 1, the maximum von Mises stress value (EQV) was concentrated in the proximal area of the primary root and cortical bone under the vertical (Figure 1a,b) and oblique loadings (Figure 1c,d). The peak EQV was higher for the oblique loading than for the vertical loading. Figure 2 shows the comparison of the distributions of the stresses on the primary molars covered with zirconia crowns under vertical loading (Figure 2a–d) and oblique 15° loading (Figure 2e–h). Comparisons of different types of finish lines showed that the feather-edged type had the highest EQV (Figure 2b,f). Figure 3 shows the comparison of the distributions of the stresses on the primary molars covered with glass-ceramic crowns under vertical (Figure 3a–d) and oblique 15° loading (Figure 3e–h). Using lithium disilicate as the other crown material had the same trend of stress distribution on the primary root (Figure 3b,f).

The loading type showed the most significant (*p* < 0.05) effect on the root (Table 2 and Table 3). The contribution percentages of the loading types were 90.25% and 99.6% for the different finish line types and materials, respectively. The cervical finish line type had a significant effect on the root (*p* < 0.05), and the contribution percentage was 8.33% (Table 2). However, the different materials of the all-ceramic crown did not have a significantly different (*p* > 0.05) effect on the root (Table 3). Tukey’s honest significant difference test indicated that the maximum EQV of the feather-edged type was significantly different (*p* < 0.05) from that of the other finish line types. Figure 4 shows the maximum EQVs of different all-ceramic materials with four types of cervical finish lines under a vertical load of 200 N. The highest value occurred on the feather-edged type. Figure 5 shows similar results under an oblique 15° load of 200 N.

## 4. Discussion

All-ceramic crowns are the newest option of dental pediatric prostheses available on the market today. These crowns represent a new option to restore the natural appearance of a child’s teeth. Children aged 4–8 years old receive a good prognosis of SSCs or zirconia crowns on posterior teeth. However, zirconia crowns are better than SSCs in the aspects of esthetics, gingival response, and plaque retention [38,39]. The importance and trend of using all zirconia crowns in children have been observed now and will expand in the future. When using all-ceramic crowns in deciduous dentition, clinicians should seriously consider tooth preparation, material properties, stress distribution, cement selection, and occlusion. Hayashi et al. [40] studied the distribution of stress on a single tooth under different loads and loading directions, and found that the maximum stress value was gathered around the alveolar crest and root curvature. Our study also revealed similar results; that is,the maximum EQV was concentrated in the proximal area of the primary root and cortical bone under vertical and oblique loading. Four cervical finish line types [27], namely, feather-edged, shoulder, mini chamfer, and chamfer, are commonly used in single-tooth restoration. For deciduous crowns, the feather-edged finish line type is usually used for SSCs. SSCs offer excellent durability and can be contoured and placed on teeth that have a limited tooth structure and where lesions extend sub-gingivally. Although they provided poor esthetics, the preparation of the cervical finish line enabled the feather-edged SSCs, which had the lowest reduction volume. However, in the present study, the feather-edged type resulted in a higher peak EQV on the primary root under different loading directions in the all-ceramic crown. Additionally, Comlekoglu et al. [27] suggested that the feather-edged type of finish line easily triggers a wedging effect at the margin, and may provide additional marginal bulk. Therefore, for all-ceramic crowns in deciduous dentition, the feather-edged type might not be recommended in clinical applications.

It was not easy to perform good moisture control for the child and chair-time restrictions using the conventional impression technique. Recently, zirconia and glass-ceramic have been successfully followed by CAD/CAM technology [41]. Additionally, Alessandretti et al. [42] suggested a completely digital workflow, from digital impressions to the final framework, with excellent patient feedback [43] and clinical reliability. Therefore, this technique seems to be particularly indicated for pediatric patients. Considering the new option of all-ceramic crowns in deciduous dentitions, the present study showed that shoulder and chamfer types with margin thicknesses of 0.4–0.6 mm appeared to have lower peak EQVs of the primary molar root. Jalalian and Aletaha [44] concluded that the chamfer and shoulder marginal designs can be used in clinical dentistry. Furthermore, they showed that the chamfer type marginal design can improve the biomechanical performance of posterior single restoration, because of the better unity and fracture resistance in the permanent crowns. However, considering that deciduous teeth are smaller than permanent teeth, and because the present study showed similar peak stress values in the shoulder and chamfer types with marginal thicknesses of 0.4–0.6 mm, clinicians should use alternatives to marginal types for single-tooth restoration. Occlusal reduction values comprise another important point to be considered when using an all-ceramic crown in the primary tooth. Regarding zirconia, an occlusal reduction of >0.8 mm is suggested for a permanent tooth prosthesis over the posterior dentition. Compared with adult dentition, deciduous teeth have a relatively low masticatory force. Therefore, a minimum occlusal space of 0.5 mm would sufficiently withstand the occlusal force, and would preserve more tooth structure providing more retention and resistance for a successful crown preparation. The operative volume of the shoulder-type cervical finish line is suggested to be 1.0 mm for permanent tooth all-ceramic crowns. Considering zirconia’s mechanical properties and the tooth structure integrity, a 0.5 mm shoulder-type or 0.4–0.6 mm chamfer-type cervical finish line is recommended for deciduous teeth to afford fracture strength and preserve the margin integrity of the prosthesis. The present study showed that the stress distribution from different materials of the all-ceramic crown did not have a significantly different effect (*p* > 0.05) on the primary root. It was an intriguing finding, as the Young’s modulus for zirconia and lithium disilicate differs quite considerably. This might be attributed to the smaller loading force (200 N) and oblique angle (15°) for children compared with adults. However, for the clinical application of glass-ceramic crowns, a 0.7–1.0 mm occlusal thickness of IPS e.max is appropriated in adult posterior dentition [24]. Therefore, a 0.7 mm occlusal reduction volume for deciduous teeth and a marginal preparation volume would be recommended for a 0.5 mm shoulder or 0.6 mm chamber margin, because of the glass-ceramic being more fragile than zirconia over the prosthesis margin.

The smaller size of the deciduous teeth underscores the importance of dental adhesive selection in ensuring the retention of indirect restorations. The two aforementioned all-ceramic crowns consist of two groups, namely: non-silicate ceramic, called zirconia, and silicate ceramic with lithium disilicate within the glass matrix. Resin cement possesses superior mechanical properties, provides excellent retention, can withstand the stresses of the oral environment, and can maintain the integrity of the tooth structure [44]. Comparing total etch-and-rinse adhesives to the self-etching adhesive system, the latter is recommended for deciduous crowns, because this would reduce the complicated and sensitive clinical step. The non-silica composition of zirconia makes it difficult to bond with the tooth structure using traditional resin composite cement. Monomers with MDP-containing resin cement (10 methacryloyloxydecyldihydrogen-phosphate) are recommended for the luting or bonding of zirconia [45,46], such as Panavia (Kuraray Medical Inc. Kurashiki, Japan) and RelyX Ultimate Clicker (3M ESPE, St. Paul, MN, USA). Moreover, silicate ceramic crowns, such as glass-ceramic, could be etched with hydrofluoric acid to enhance the bonding strength; clinicians could thus choose self-etching adhesives, such as Panavia, or total etch-and-rinse adhesives, such as Variolink II (Ivoclar Vivadent, Schaan, and Liechtenstein), depending on the patient.

Regarding occlusal adjustment and polishing, all-ceramic crowns require more adjustment steps than SSCs, based on their material property. However, an inaccurate interdental space estimation can prolong the clinical operative time and might increase stress on the roots due to malocclusion. High occlusion contact points also easily cause crown exfoliation, even when a dental adhesive with a high bonding-strength is used. All-ceramic crowns are harder than SSCs and natural teeth, and surface polishing after occlusal adjustment is more important in all-ceramic crowns than in SSCs. A prosthesis with a rough surface and high hardness would cause a wearing of the antagonistic teeth. Postoperative follow-up after one week is recommended to monitor the gingival and occlusion condition.

The limitations of this study included a lack of validation with experiments and/or other types of investigations. For the results from the FEM, we provided the qualitative results for the four finish types and added the computational effort based on the statistical methods. Our study presented the ideal finish line preparation when fabricating pediatric all-ceramic crowns. Children’s unpredictable behavior often poses a great challenge for dentists during treatment. With the help of modern digital technology and the advancement of dental material, dentists can achieve optimum precision in all-ceramic crown fabrication, while also achieving a great prognosis for the child. Having a great dental experience early on will enhance the dentist–patient relationship, which is beneficial to both dentists and the patients.

## 5. Conclusions

The maximum EQVs of the primary roots of four different types of finish lines (shoulder 0.5 mm, feather-edged, chamfer 0.6 mm, and mini chamfer 0.4 mm) and two all-ceramic crown materials (zirconia and lithium disilicate) were studied using FEA. The results revealed that the peak stress value was concentrated in the proximal area of the primary root and cortical bone. Oblique loading showed higher values than those from the vertical loading. In addition, the shoulder-type finish line with a 0.5 mm margin thickness and the chamfer-type finish line with a 0.4–0.6-mm thickness showed lower peak stress values on the roots than the feather-edged type. Lastly, zirconia and lithium disilicate showed no significant difference in stress distribution under 200 N loading, with correct bonding between the prosthesis and primary tooth. However, clinicians should still focus on tooth preparation volume, occlusion adjustment, and follow-up.

## Figures and Tables

**Figure 1 materials-13-01094-f001:**
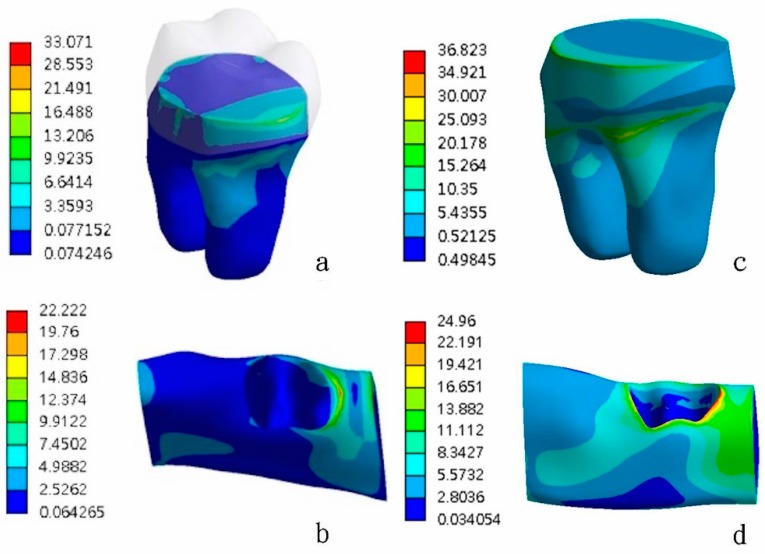
Distributions of the stresses on a primary molar covered with an all-ceramic crown. The maximum von Mises stress value (EQV) was concentrated in the proximal area of the primary root and cortical bone under vertical loading (**a**,**b**) and oblique loading (**c**,**d**).

**Figure 2 materials-13-01094-f002:**
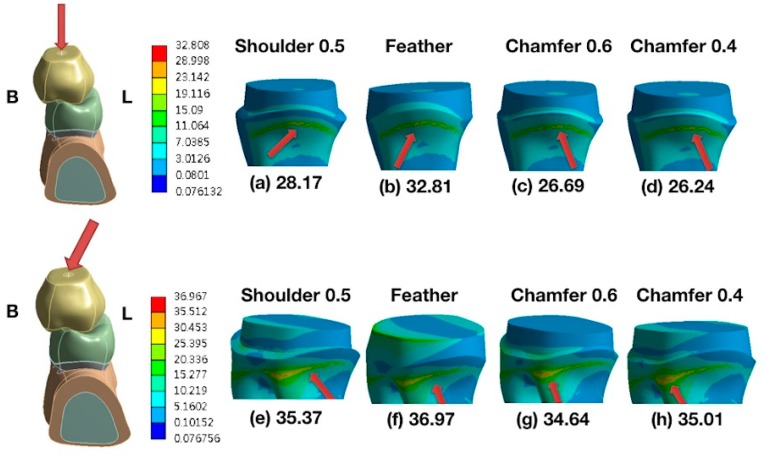
Distributions of the stresses on primary molars covered with zirconia crowns. (Top) A vertical load of 200 N applied to the lower, primary second molars. From left to right, the loading models are as follows: (**a**) shoulder (0.5 mm), (**b**) feather-edged, (**c**) chamfer (0.6 mm), and (**d**) mini chamfer (0.4 mm). (Bottom) The oblique 15° load of 200 N applied to the lower, primary second molars. From left to right, the loading models are as follows: (**e**) shoulder (0.5 mm), (**f**) feather-edged, (**g**) chamfer (0.6 mm), and (**h**) mini chamfer (0.4 mm). Red arrows indicate the peak stresses and locations. B—buccal; L—lingual.

**Figure 3 materials-13-01094-f003:**
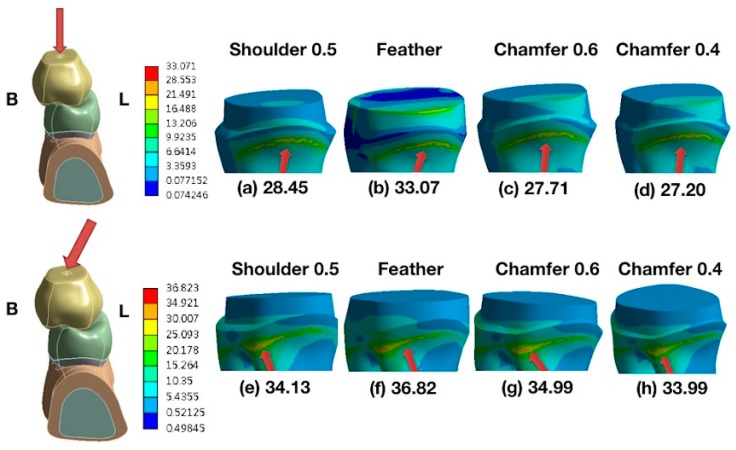
Distributions of stresses on primary molars covered with glass-ceramic crowns. (Top) A vertical load of 200 N applied to the lower, primary second molars. From left to right, the loading models are as follows: (**a**) shoulder (0.5 mm), (**b**) feather-edged, (**c**) chamfer (0.6 mm), and (**d**) mini chamfer (0.4 mm). (Bottom) The oblique 15° load of 200 N applied to the lower, primary second molars. From left to right, the loading models are as follows: (**e**) shoulder (0.5 mm), (**f**) feather-edged, (**g**) chamfer (0.6 mm), and (**h**) mini chamfer (0.4 mm). The red arrows indicate the peak stresses and locations. B—buccal; L—lingual.

**Figure 4 materials-13-01094-f004:**
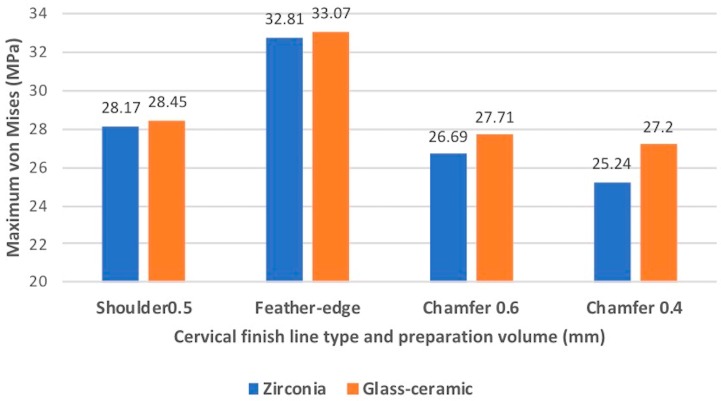
Comparison of the maximum EQVs of the all-ceramic crowns with four types of cervical finish lines under a vertical load of 200 N. The highest value occurred on the feather-edged type.

**Figure 5 materials-13-01094-f005:**
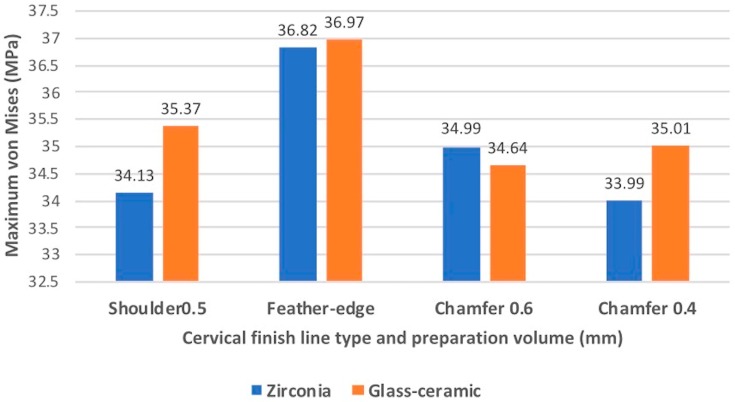
Comparison of the maximum EQVs of the all-ceramic crowns with four types of cervical finish lines under an oblique 15° load of 200 N. The highest value occurred on the feather-edged type.

**Table 1 materials-13-01094-t001:** Materials and properties of the finite element models.

Material	Young’s Modulus (GPa)	Poisson Ratio (ν)
ZrO_2_ [29]	220	0.3
Lithium disilicate [20]	91	0.23
Cortical bone [30]	13.7	0.3
Cancellous bone [30]	1.85	0.3
Dentin [31]	18.6	0.31
Resin cement [32]	8.3	0.3

**Table 2 materials-13-01094-t002:** Summary of the analysis of variance (ANOVA) showing the statistical results of the maximum stresses of different loading and finish line types concerning the primary molar.

Source	DF	SS	MS	%TSS	P
Loading type	1	149.19	149.19	90.25	0.000*
Finish line type	3	41.286	13.76	8.33	0.000*
Finish line type x Loading type	3	7.01	2.34	1.42	0.002*
Total		165.29		100.00	

*Significantly different at *p* < 0.05; DF—degrees of freedom; SS—sum of squares; MS—mean of squares; TSS—total sum of squares. Loading type: vertical and oblique load (15° from the tooth’s long axis). Finish line type: shoulder (0.5 mm), feather-edged, chamfer (0.6 mm), and mini chamfer (0.4 mm).

**Table 3 materials-13-01094-t003:** Summary of the ANOVA showing the statistical results of the maximum stresses of different loading and ceramic types concerning the primary molar.

Source	DF	SS	MS	%TSS	P
Loading type	1	149.19	149.19	99.6	0.000*
Ceramic type	1	0.31	0.31	0.2	0.79
Loading type x Ceramic type	1	0.22	0.22	0.2	0.82
Total		149.72		100.00	

*Significantly different at *p* < 0.05. DF—degrees of freedom; SS—sum of squares; MS—mean of squares; TSS—total sum of squares. Loading type: vertical and oblique load (15° from the tooth’s long axis). Ceramic type: zirconia and glass-ceramic.

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
