# Peer review of "Comparison of Different Cervical Finish Lines of All-Ceramic Crowns on Primary Molars in Finite Element Analysis"

_materials, 2020, doi:10.3390/ma13051094_

Round 1
Reviewer 1 Report
The manuscript is well written and clear. It has a straight forward structure and a detailed objective. The concluions are substained by the data described in the work.
However, some minnor information is required to completely follow the manuscript. Why did the authors selected von Mises stress values for mechanical testing evaluation?
Moreover, Figure 4 shows the comparison of the mean EQVs of all-ceramic crowns with four types of cervical finish lines under vertical and oblique loading. However, it is not clear if the data shown came ftom the vertical or the oblique loading or is an average of both.
Reviewer 2 Report
Overall. General English grammar revision (Minor spelling errors).
Abstract: ok
Introduction. Authors stated “Low-level caries of primary teeth could be treated by caries removal, cavity preparation, and cavity restoration using materials such as composite resins and glass ionomers”. Please add a reference for this statement.
Introduction. Authors stated “The traditional SSC is sufficiently flexible to allow trimming, crimping, and shaping for better fit to and retention by the primary teeth cervical area”. Please define SSC before using the abbreviation.
Materials and Methods. Authors stated “The applied vertical load had a magnitude of 200 N, and the applied oblique force of 200 N was set at 15° from the long axis of the tooth”. Please add a reference for this Method.
Materials and Methods. Authors stated that “Data were compared using analysis of variance”. Please specify the kind of analysis used pointing out if parametric (example: ANOVA) or non parametric (example: Kruskal Wallis…) test was used. Additionally, please point out how normality of distributions was assessed (Kolmogorov Smirnov? Shapiro Wilk?).
Materials and Methods. Analysis of variance, if significant, should be followed by post hoc analysis: for example Tukey test would be performed for Gaussian distributions, alternatively Mann Whitney for non Gaussian data.
Materials and Methods. Authors stated. “All statistical analyses were conducted using SPSS (Version 20c.0 for Windows)”. Please point our manufacturer, city and State.
Materials and Methods. Please point out significance level of statistical tests (P<0.05?).
Discussion. More discussion could be added considering the opportunity of a ceramic restoration in paediatric patients. In fact, rigidity could be very high, and costs could be high too. Additionally, the age of the patient and the age of esteemed permutation have to be taken into careful account. These aspects have to be considered before planning a zirconia crown in primary teeth.
Discussion. Authors could stress more the importance of zirconia crowns and their adaptability to a complete digital workflow. It could be stated that “Zirconia can be successfully faced with CAD/CAM technology, thus allowing a completely digital workflow, from impression to final framework, with clinical reliability (Reliability and mode of failure of bonded monolithic and multilayer ceramics. Alessandretti R, Borba M, Benetti P, Corazza PH, Ribeiro R, Della Bona A. Dent Mater. 2017 Feb;33(2):191-197) and excellent patients feedback (Computerized Casts for Orthodontic Purpose Using Powder-Free Intraoral Scanners: Accuracy, Execution Time, and Patient Feedback. Sfondrini MF, Gandini P, Malfatto M, Di Corato F, Trovati F, Scribante A. Biomed Res Int. 2018 Apr 23;2018:4103232.). Therefore, this technique seem to be particularly indicated for paediatric patients”.
References. Some old references have been presented (1950, 1950, 1989, 1994, 1975, 1998). Where possible, please change with some modern research. Some fresher references have been suggested in Discussion Section.
Figure 4: Please point out the meaning of error bars (standard deviation? Standard Error? Confidence Interval?).
Tables: A table with the results of descriptive statistics could be added. Mean, Standard Deviation, Minimum, Median and Maximum values could be reported, in order to help the readers in the interpretation of the results.
Reviewer 3 Report
Dear Authors,
After the review process, I have several comments: you should rewrite the Materials and Methods, you should divide it in separate sections with relevant references; you should insert a limitation of the study; you should present a future valorization of the research.
Best regards!
Reviewer 4 Report
The manuscript “Comparison Different Cervical Finish Line of All-Ceramic Crowns on Primary Molars in FEA” looks interesting, but some improvements must be made before to be suitable for publication. The comments related to each manuscript section could be find below.
Introduction
Line 25 “Conclusion is the shoulder…” - Please check the sentence.
Line 46 “Computer-aided design/Computer-aided manufacturing” instead of “computerized-aided design/computerized-aided manufacture”
Line 55 “Young’s modulus” instead of “Young’s module”
Line 71 “equivalent forces of” instead of “forces of”
Line 78 Consider reformulating
Line 85 “made of” instead of “made with” and “using finite element analysis” instead of “in finite element analysis”
The section Materials and Methods is very poor described.
There is no information on geometrical model, how many components there are, and what are the contacts between those components. The mesh statistics (nodes and elements) does not reflect the individual components but rather a summation. Also, there is no information regarding the FEA software.
The loading directions, values and points/surfaces of application are not clearly presented and correlated with the physiology. The influence of those on the results is very significant.
Applying statistics to the simulation results is a poor practice since the finite element method is by definition an approximate method of solving the real problem.
It is mandatory to confirm the simulation results with experiments and/or other types of investigation (microscopy).
Results
The stress results are from the surface of the model? How about the interior? Why Von Mises stress is considered, since the literature present results of shear stress for the large majority of dental materials.
The Degrees of freedom in ANOVA indicate very few data results taken into consideration. Too few to have a statistical significance however. The data have only lower standard deviation (figure 4)? Why?
Discussions
The discussions are related to a series of simulation data that are not to be trusted in this phase of the study. It is highly recommended to reconsider the study from the beginning. The idea may present some interest but the soundness of the research is not high enough.
Round 2
Reviewer 2 Report
Good job
Reviewer 4 Report
Authors perform many modifications who improve the quality of the paper.
I still suggest to reformulate the conclusion, without bullets, and especially the sentence "Zirconia and lithium disilicate showed no significant difference in stress distribution; however, clinicians should still focus on cement selection, occlusion adjustment, and follow-up." Because these materials have different Young modulus, why this fact didn't affect the stress distribution?
